# Inhibition of PRMT5/MEP50 Arginine Methyltransferase Activity Causes Cancer Vulnerability in NDRG2^low^ Adult T-Cell Leukemia/Lymphoma

**DOI:** 10.3390/ijms25052842

**Published:** 2024-02-29

**Authors:** Tomonaga Ichikawa, Akira Suekane, Shingo Nakahata, Hidekatsu Iha, Kazuya Shimoda, Takashi Murakami, Kazuhiro Morishita

**Affiliations:** 1Division of Tumor and Cellular Biochemistry, Department of Medical Sciences, University of Miyazaki, Miyazaki 889-1692, Japan; snakahata@kufm.kagoshima-u.ac.jp (S.N.); kmorishi@med.miyazaki-u.ac.jp (K.M.); 2Department of Microbiology, Saitama Medical University, 38 Morohongo, Moroyama, Iruma-gun, Saitama 350-0495, Japan; takmu@saitama-med.ac.jp; 3Trauma and Acute Critical Care Center, Tokyo Medical and Dental University Hospital, Tokyo 113-8510, Japan; akira.suekane0304@gmail.com; 4Division of HTLV-1/ATL Carcinogenesis and Therapeutics, Joint Research Center for Human Retrovirus Infection, Kagoshima University, Kagoshima 890-8544, Japan; 5Division of Pathophysiology, The Research Center for GLOBAL and LOCAL Infectious Diseases (RCGLID), Oita University, Yufu 879-5503, Japan; hiha@oita-u.ac.jp; 6Division of Hematology, Diabetes, and Endocrinology, Department of Internal Medicine, Faculty of Medicine, University of Miyazaki, Miyazaki 889-1692, Japan; kshimoda@medmiyazaki-u.ac.jp; 7Project for Advanced Medical Research and Development, Project Research Division, Frontier Science Research Center, University of Miyazaki, Miyazaki 889-1692, Japan

**Keywords:** NDRG2, PRMT5, MEP50, cancer vulnerability, ATL

## Abstract

N-myc downstream-regulated gene 2 (NDRG2), which is a tumour suppressor, is frequently lost in many types of tumours, including adult T-cell leukaemia/lymphoma (ATL). The downregulation of NDRG2 expression is involved in tumour progression through the aberrant phosphorylation of several important signalling molecules. We observed that the downregulation of NDRG2 induced the translocation of protein arginine methyltransferase 5 (PRMT5) from the nucleus to the cytoplasm via the increased phosphorylation of PRMT5 at Serine 335. In NDRG2^low^ ATL, cytoplasmic PRMT5 enhanced HSP90A chaperone activity via arginine methylation, leading to tumour progression and the maintenance of oncogenic client proteins. Therefore, we examined whether the inhibition of PRMT5 activity is a drug target in NDRG2^low^ tumours. The knockdown of PRMT5 and binding partner methylsome protein 50 (MEP50) expression significantly demonstrated the suppression of cell proliferation via the degradation of AKT and NEMO in NDRG2^low^ ATL cells, whereas NDRG2-expressing cells did not impair the stability of client proteins. We suggest that the relationship between PRMT5/MEP50 and the downregulation of NDRG2 may exhibit a novel vulnerability and a therapeutic target. Treatment with the PRMT5-specific inhibitors CMP5 and HLCL61 was more sensitive in NDRG2^low^ cancer cells than in NDRG2-expressing cells via the inhibition of HSP90 arginine methylation, along with the degradation of client proteins. Thus, interference with PRMT5 activity has become a feasible and effective strategy for promoting cancer vulnerability in NDRG2^low^ ATL.

## 1. Introduction

Adult T-cell leukaemia/lymphoma (ATL) is a highly aggressive malignancy caused by infection with the oncogenic retrovirus human T-cell leukaemia virus type 1 (HTLV-1) in CD4^+^ T-lymphocytes. While the incidence of new HTLV-1-infected carriers and ATL is decreasing in Japan, the presence of HTLV-1 carriers and associated diseases indicates an increasing tendency in endemic and non-endemic areas [1,2]. Conventional chemotherapies have significantly improved the survival of ATL patients; however, the majority of ATL is resistant to chemotherapeutic agents, and there are few satisfactory treatment options for relapsed refractory ATL with poor prognosis. Moreover, the detailed mechanism of ATL progression has not been completely elucidated, and few molecular-targeted therapeutic agents have been clinically approved [3,4].

Recently, we reported that the tumour suppressor gene N-myc downstream-regulated gene 2 (NDRG2) was significantly downregulated through the accumulation of genetic and epigenetic abnormalities, which is involved in tumour incidence, progression, and metastasis in many types of tumours, including ATL, oral, pancreatic, liver, or other tumours [5,6,7]. Because NDRG2 is associated with the dephosphorylation of PTEN at Serine 380, Threonine 382, and Threonine 383 (STT) in its C-terminal domain via the recruitment of Serine/Threonine Protein Phosphatase 2A (PP2A), the downregulation of NDRG2 expression plays an important role in the development of disease in HTLV-1-infected and ATL cells through the aberrant activation of PI3K/AKT and NF-κB signal transduction pathways; this effect is accomplished via the functional inactivation of high-phosphorylated PTEN [8,9]. Moreover, to seek the precise molecular mechanism of tumour development through the inactivation of NDRG2, we profiled phosphopeptides regulated by NDRG2/PP2A via 2-dimensional image converted analysis of liquid chromatography-mass spectrometry (2-DICAL). We identified protein arginine methyltransferase 5 (PRMT5) as a NDRG2/PP2A-modulating central signalling molecule, and we observed that PRMT5 was translocated from the nucleus to the cytoplasm through the increased phosphorylation of Serine 335 in NDRG2^low^ ATL cells [10]. Furthermore, it has been reported that PKC-mediated phosphorylation of PRMT5 at Serine 15 enhanced the methyltransferase activity and induced the NF-κB signalling pathway through the interaction of PRMT5 with p65 in colorectal cancer [11], and the inhibition of myosin phosphatase (MP) activity resulted in the high phosphorylation of PRMT5 at Threonine 80 along with the arginine methylation of histones and gene repression in hepatocellular carcinoma [12]. We suggest that the phosphorylation of PRMT5 is critical for enzymatic activity and tumour development.

Arginine methylation is a posttranslational modification catalysed by members of the protein arginine methyltransferase (PRMT) family. Of the three types of methylation products, PRMT5 is a type II PRMT that generates mono- and symmetric dimethyl arginine (SDMA) modifications on protein substrates through the utilisation of S-adenosylmethionine (SAM) as the methyl donor; in addition, it is involved in organelle biogenesis, epigenetic remodelling, maintenance of haematopoietic stem cells, and tumorigenesis via arginine methylation of histone and non-histone proteins [13,14]. Moreover, the activity and substrate specificity of PRMT5 is regulated by posttranslational modifications and adaptor proteins such as RioK1, pICIn, and COPR5 [15,16,17]. Among them, the PRMT5 complex with methylosome protein 50 (MEP50) forms a hetero-octamer, which stimulates enzymatic arginine methyltransferase activity [18]. PRMT5/MEP50 catalyses arginine methylation of histone H3/H4 and non-histone proteins, such as transcription factor p53, to induce the modulation of chromatin structure and gene silencing in the nucleus [19,20]. Furthermore, it has been reported that PRMT5 and MEP50 expression is upregulated in various cancers, and the cytoplasmic localisation of PRMT5/MEP50 is associated with a wide variety of cellular processes, including signal transduction pathways that are highly relevant to the pathogenesis of cancer [21,22,23]. We demonstrated that PRMT5 was localised in the nucleus with high SDMA on histone 3 (H3R8me2s) and histone 4 (H4R3me2s) in NDRG2-expressing cells, whereas the translocation of phosphorylated PRMT5 from the nucleus to the cytoplasm in NDRG2^low^ ATL cells enhanced the binding with HSP90A. These resulted in the maintenance of chaperone activity and oncogenic client proteins through arginine methylation of HSP90A (Arginine 345 and 386). Furthermore, the knockdown of PRMT5 expression and HSP90A mutants with arginine substitution induced apoptosis through the degradation of client proteins with the loss of HSP90A arginine methylation [10]. These demonstrated that arginine methylation of HSP90 through the phosphorylation of PRMT5 in the cytoplasm plays an important role in tumour progression and may represent a therapeutic target.

In this study, the inhibition of PRMT5/MEP50 arginine methyltransferase activity with specific short hairpin RNA (shRNA)-induced knockdown or treatment with the SAM-competitive PRMT5-specific inhibitors CMP5 and HLCL61 significantly suppressed cell proliferation via the degradation of client proteins AKT and NEMO in NDRG2^low^ ATL and many other cancer cells. This effect was elicited via a decrease in arginine methylation of HSP90A. The suppression of PRMT5/MEP50 activity in NDRG2-expressing cells did not affect the suppression of cell proliferation and the expression of HSP90 client proteins. Moreover, the knockdown of NDRG2 with shRNA in NDRG2-expressing cells exhibited an increase in HSP90A arginine methylation, along with a decrease in H3R8me2s/H4R3me2s; additionally, it demonstrated the vulnerability to treatment with PRMT5 inhibitors through the degradation of AKT and NEMO. Furthermore, the enhanced expression of NDRG2 remarkably inhibited the suppression of cell proliferation and the protein degradation of AKT and NEMO with the treatment of PRMT5 inhibitors in tumour cells. Because PRMT5/MEP50 inhibition significantly induced the suppression of cell proliferation and client proteins in leukaemic cells from ATL patients, we herein propose that interference with cytoplasmic PRMT5/MEP50 is a feasible and effective strategy for promoting cancer vulnerability in NDRG2^low^ ATL and various cancer cells.

## 2. Results

### 2.1. The Knockdown of PRMT5/MEP50 Expression Results in the Inhibition of Cell Proliferation through the Degradation of Client Proteins in ATL and Various Cancer Cells with Low NDRG2 Expression

The tumour suppressor gene NDRG2 is a PP2A phosphatase recruiter and regulates PRMT5 phosphorylation. Additionally, PRMT5 in NDRG2^low^ ATL and cancer cells is hyperphosphorylated and translocated to the cytoplasm, where it binds to HSP90, thus contributing to the maintenance of a high function of its client proteins. PRMT5 acts as a heterodimer of the nonenzymatic cofactor MEP50, which is an obligate partner and is required for arginine methyltransferase activity [18]. To investigate the function of the PRMT5/MEP50 heterodimer in NDRG2-downregulated cells, we separately suppressed their expression and examined their effects on HSP90 function in low NDRG2-expressing ATL cell lines (KK1 and SO4) and the high NDRG2-expressing cell lines T cell acute lymphoblastic leukaemia T-ALL (Jurkat and MOLT4).

We examined the effect of PRMT5 knockdown by using shRNA with two different sequences against PRMT5 in low NDRG2-expressing ATL cell lines. Although MEP50 expression was unaffected by PRMT5 knockdown, the cell proliferation rate and HSP90 client proteins (AKT and NEMO) were significantly reduced in the two PRMT5-knockdown ATL cell lines compared with parental and shRNA against luciferase (shluc) as a negative control (Figure 1A,B and Appendix A). Furthermore, ATL cell lines remarkably induced the suppression of the cell growth and protein levels of AKT and NEMO through the downregulation of MEP50 expression without the effect of PRMT5 expression (Figure 1C,D and Appendix A). As a control, the suppression of PRMT5 or MEP50 expression in the NDRG2-expressing T-ALL cell lines exhibited limited effects on cell growth inhibition and no degradation of HSP90 client proteins (Figure 1E–H and Appendix A), suggesting that the effects of PRMT5/MEP50 activity might be dependent on the expression of NDRG2.

We previously observed that the expression of NDRG2 was low in many types of solid cancers, including SAS (oral squamous cell carcinoma) and U2OS (osteosarcoma), and cell proliferation was significantly suppressed by silencing PRMT5 expression with the degradation of AKT and NEMO proteins [10]. Therefore, we introduced a shRNA-expressing construct for MEP50 in two solid cancer cell lines to examine the degradation of HSP90 client proteins. As a result, the inhibition of MEP50 expression in solid tumour cell lines decreased the signalling molecules AKT and NEMO as HSP90 client proteins (Appendix A).

### 2.2. Hyperphosphorylated PRMT5 Binds to MEP50 and Promotes HSP90 Arginine Methylation

Since PRMT5 directly binds with HSP90 along with the increase in HSP90 chaperone activity [10], we examined whether MEP50 is important for the binding of PRMT5 and HSP90. HA-tagged HSP90, EGFP-tagged PRMT5, and/or Flag-tagged MEP50 were cotransfected into 293T cells in different combinations, and the protein complex was precipitated with anti-HA (HSP90) or anti-GFP (PRMT5) antibodies. HSP90 could be detected in the protein complex from 293T cells transfected with the PRMT5-expressing vector, and the protein complex between HSP90 and PRMT5 was enhanced via transfection with the MEP50-expressing vector, followed by an increase in HSP90 arginine methylation that was recognised by using the SYM10 antibody (Figure 2A). To investigate whether MEP50 was directly associated with HSP90, HA-tagged HSP90, Flag-tagged MEP50, and/or EGFP-tagged PRMT5 transiently expressed in 293T cells were purified by immunoprecipitation with anti-HA (HSP90) or anti-Flag (MEP50) antibodies. The binding was hardly observed when MEP50 alone was expressed with HSP90. However, when MEP50 was expressed with PRMT5, it bound to HSP90 and promoted the arginine methylation reaction of HSP90 (Appendix A). Therefore, we indicated that the binding of HSP90 with PRMT5 enhanced the arginine methylation of HSP90 through the presence of MEP50. Furthermore, to examine whether the arginine methyltransferase activity of PRMT5 is important for MEP50 binding, we used an EGFP-tagged PRMT5 mutant with G367A/R368A (GR/AA), which lost its enzyme activity [24]. Immunoprecipitation from cells transfected with PRMT5/WT or GR/AA did not change their binding ability to MEP50 (Appendix A). We previously identified that the phosphorylation of PRMT5 at S335 plays a pivotal role in enzyme activity and localisation into the cytoplasm [10], and we further determined the interaction between PRMT5 with a mutant with alanine (with S335A as the dephosphorylation statue) or aspartate (with S335D as the high phosphorylation statue) and MEP50. MEP50 coprecipitated PRMT5/WT or S335D; however, the immunoprecipitate from S335A reduced its binding ability to MEP50 (Figure 2B), indicating that MEP50 firmly associated with the phosphorylated PRMT5 at S335, which resulted in the enhancement of HSP90 arginine methylation. These results suggest that the complex with PRMT5 and MEP50 in the cytoplasm plays an important role in cancer development and that the inhibition of PRMT5/MEP50 activity exhibits anti-tumour effects regarding cancer vulnerability only in NDRG2^low^ cancer cells.

### 2.3. NDRG2^low^ ATL Cells Are Sensitive to PRMT5 Inhibitors

To confirm the relationship of cancer vulnerability between low NDRG2 expression and PRMT5/MEP50 activity, we used three recently developed potent and specific PRMT5 inhibitors (EPZ015866, CMP5, and HLCL61). Initially, we used EPZ015866, which potently inhibits the protein substrate-binding pocket, followed by the suppression of histone arginine methylation in mantle cell lymphoma (MCL) and lung cancer in the nucleus. EPZ15866 was synthesised by structure and property-guided design strategies using EPZ015666 [25,26,27]. Although EPZ015866 was administered to ATL and T-ALL cell lines at a high concentration of 100 μM, cell growth inhibition did not reach 50% inhibition, resulting in the discontinuation of EPZ015866 use (Table 1). Subsequently, cell growth inhibition experiments using two HTLV-1-infected cell lines (HTLV-1 oncogenic protein Tax-positive MT2 and HUT102), five ATL cell lines (Tax-positive KOB and SU9T-01, Tax-negative KK1, SO4, and ED), and three T-ALL cell lines (Jurkat, MOLT4, and MKB1) were performed via two SAM-competitive inhibitors (CMP5 and HLCL61), which selectively and reversibly bind to the active site of PRMT5 and inhibit PRMT5-mediated arginine methylation [28,29]. All of the HTLV-1-infected and ATL cell lines were sensitive to CMP5 compared to T-ALL cell lines, with IC50 values ranging from 3.98 to 21.65 μM for ATL-related cell lines and 32.5–92.97 μM at 120 h for T-ALL cell lines. Furthermore, HLCL61 was more sensitive to ATL-related cell lines than to T-ALL cell lines, with IC50 values ranging from 3.09 to 7.58 μM at 120 h for ATL-related cell lines and 13.06–22.72 μM for T-ALL cell lines (Figure 3A and Table 1). The effect of PRMT5 inhibitors has also been observed in NDRG2-expressing T-ALL cell lines, and PRMT5 is present in the cytoplasm and the nucleus, followed by the detection of H3R8me2s/H4R3me2s and the maintenance of physiological functions [10,13,14]. Therefore, we examined the inhibitory effect of PRMT5 inhibitors on histone arginine methylation in two T-ALL cell lines and found that the expression of H3R8me2s/H4R3me2s was decreased in a concentration-dependent manner (Appendix A). We have reported that PRMT5 in NDRG2^low^ ATL cell lines is mainly localised to the cytoplasm and regulated HSP90 chaperone function rather than arginine methylation of histones. Moreover, protein expression levels of PRMT5 and HSP90 were unchanged in ATL and T-ALL cell lines after treatment with PRMT5 inhibitor; however, HLCL61 treatment markedly reduced both AKT and NEMO expression, which are known client proteins of HSP90 in a dose-dependent manner. Furthermore, the use of inhibitors on the control T-ALL cell lines did not alter the protein expression levels of AKT and NEMO (Figure 3B,C). In addition, after treatment with PRMT5 inhibitor, HSP90 in ATL-related cell lines was precipitated by specific antibodies; in addition, bound PRMT5 and the level of arginine methylation in HSP90 were examined via western blots by the use of each specific antibody. PRMT5 binding to HSP90 and methylated arginine in HSP90 were reduced in HTLV-1-infected and ATL cell lines upon treatment with PRMT5 inhibitor (Figure 3D), indicating that PRMT5 inhibitors show the inhibition of cell proliferation via the suppression of HSP90 activity in NDRG2^low^ ATL, but not in T-ALL.

### 2.4. Knockdown of NDRG2 Expression in T-ALL Cells Enhances Sensitivity to PRMT5 Inhibitors

To examine changes in PRMT5/MEP50 function depending on the expression level of NDRG2, we performed experiments using NDRG2-specific shRNA to suppress NDRG2 expression in T-ALL cell lines. The inhibition of NDRG2 expression in T-ALL cell lines exhibited suppressed histone arginine methylation, increased binding of HSP90 to PRMT5, and increased arginine methylation of HSP90. In contrast, the transduction of shluc showed higher histone arginine methylation and little binding of HSP90 to PRMT5, as was observed in the parental T-ALL cells (Figure 4A). As a result, the NDRG2 low-expressing T-ALL cell lines exhibited significantly decreased expression of AKT and NEMO, with cell growth suppressed by treatment with the PRMT5 inhibitors HLCL61 or CMP5 compared with the parental and shluc T-ALL cell lines (Figure 4B–D and Appendix A). To further examine the role of PRMT5 in mouse embryonic fibroblasts (MEFs) of *Ndrg2*-deficient (−/−), we investigated the expression of histone arginine methylation in wild-type (WT, +/+) and *Ndrg2*(−/−) MEFs. The detection of H4R3me2s was remarkably reduced in *Ndrg2*(−/−) compared with WT(+/+) MEFs without the effect of PRMT5 expression (Appendix A). Furthermore, the knockdown of PRMT5 expression significantly decreased AKT and NEMO expression in *Ndrg2*(−/−) MEFs but not in WT(+/+) MEFs (Figure 4E). These results indicate that the reduced expression of NDRG2 translocates PRMT5 from the nucleus to the cytoplasm and supports the function of HSP90 in the cytoplasm. Of note, the decreased expression of NDRG2 also enhanced the growth inhibitory function of PRMT5 inhibitors.

### 2.5. The Enhanced Expression of NDRG2 Attenuates the Antitumour Effects of PRMT5 Inhibitors in ATL and Solid Cancer Cells

In the next experiment, NDRG2 was transfected into ATL and solid cancer cell lines, and PRMT5 inhibitors were administered to examine their effects. As expected, the introduction of NDRG2 expression indicated a lower sensitivity to PRMT5 inhibitors compared with parental and Mock as a negative control ATL (Figure 5A and Appendix A). Furthermore, the treatment of the PRMT5 inhibitor did not induce protein degradation of AKT and NEMO in ATL cell lines with the ectopic expression of NDRG2 (Figure 5B,C). We also examined the effects of PRMT5 in solid cancer cell lines, and the enhanced expression of NDRG2 suppressed the inhibitory effect of cell proliferation and the degradation of client proteins by the treatment of PRMT5 inhibitor HLCL61 (Appendix A). These results indicate that PRMT5 inhibition is selective and potent to NDRG2^low^ ATL and various cancer cells.

### 2.6. PRMT5 Inhibitors Are Effective against NDRG2^low^ ATL Patient Cells

We further analysed the cytotoxic effects of PRMT5 inhibitors in primary cells from acute-type ATL patients. Although some ATL cells from the patients are resistant to PRMT5 inhibitors, the majority of ATL patient cells were more sensitive to PRMT5 inhibitors than Peripheral blood mononuclear cells (PBMCs) as healthy control, with IC50 values of 23.94–33.12 μM at 120 h for ATL patients versus PBMCs of 58.08 μM in CMP5, as well as IC50 values of 2.33–42.71 μM at 120 h for ATL patients versus PBMCs of 43.37 μM in HLCL61 (Table 2). Significant differences between normal PBMCs and ATL patient samples were calculated with the treatment of CMP5 and HLCL61 at 50 μM and 20 μM, respectively (Figure 6A). Moreover, NDRG2 protein expression in ATL patient cells was remarkably lower than that in CD4^+^ T-cells. The expression of PRMT5 and MEP50 was similar to CD4^+^ T-cells (Figure 6B), suggesting that the sensitivity to PRMT5 inhibitors may be inversely proportional to NDRG2 expression. Furthermore, the expression of AKT and NEMO proteins was decreased in those ATL cells in a dose-dependent manner (Figure 6C,D), thus indicating that PRMT5/MEP50 inhibition can be a promising therapeutic target for cancer vulnerability in NDRG2^low^ ATL cells.

## 3. Discussion

A novel tumour suppressor gene, NDRG2, is downregulated in almost all cases of ATL, as well as in the majority of various cancers, such as liver, lung, colorectal, oral, and brain tumours. The decreased expression of NDRG2 induces the disruption of the homeostatic mechanism of the stress response, chronic inflammation, and aberrant activation of signal transduction pathways through increased phosphorylation of important signalling molecules; moreover, it is associated with tumour development, poor prognosis, metastasis, drug resistance, and a decreased survival rate [30,31]. The analysis of NDRG2 function, followed by the identification of therapeutic targets, is a significant clinical need [32]. Therefore, it is likely that an inhibitor that specifically induces synthetic lethality in NDRG2^low^ cancer cells may have a cancer-specific effect and avoid the side effects in normal cells that maintain NDRG2 expression. We profiled molecular partners of NDRG2 to address this challenge and identified PRMT5, which regulates arginine methylation of several molecules in both the nucleus and the cytoplasm to participate in several physiological processes and tumorigenesis. We found that dephosphorylated PRMT5 via the recruitment of NDRG2/PP2A was relocated to the nucleus in NDRG2-expressing cells, displaying an association with arginine methylation of histone. Therefore, the treatment of PRMT5-specific inhibitors did not affect the expression of AKT and NEMO despite the inhibition of histone arginine methylation (Figure 7A). Highly phosphorylated PRMT5 was tightly complexed with MEP50 in the cytoplasm, followed by the enhancement of HSP90 arginine methylation in NDRG2^low^ ATL and other cancer cells. Thus, the targeting of PRMT5/MEP50 activity significantly induced the suppression of cell growth and the degradation of client proteins AKT and NEMO in NDRG2^low^ ATL and other cancer cells (Figure 7B). We identify a novel vulnerability in ATL resulting from the relationship of the synthetic lethality between the loss of NDRG2 expression and cytoplasmic PRMT5/MEP50 activity and reveal a functional and promising therapeutic target for the treatment of NDRG2^low^ tumours.

Synthetic lethal therapy has been explored to target many cancers with dysregulated or mutated genes, ranging from oncogenes to tumour suppressors and factors related to DNA repair, metabolism, and genetic background; in addition, this therapy can promote the indirect targeting of mutations by identifying alternative molecules [33,34]. It has been reported that the poly (ADP-ribose) polymerase (PARP) inhibitor olaparib, which is involved in the DNA damage response, is clinically used for many types of solid cancer induced by mutations in the tumour suppressor gene BRCA1/2 [35,36]. Furthermore, the relationship between p53-mutant and ATR/Chk [37,38], or PTEN-deficient and ATM/PARP [39,40], are associated with synthetic lethality leading to the development of anticancer-targeted therapies. Because the combination with the downregulation of tumour suppressors and genes involved in cell proliferation is a candidate for synthetic lethality, cytoplasmic PRMT5/MEP50 may be a therapeutic target in NDRG2^low^ cancers.

PRMT5 forms a hetero-octomer with the nonenzymatic molecule MEP50, which exhibits arginine methylation of histones and non-histone proteins in both the nucleus and the cytoplasm through the enhancement of arginine methyltransferase activity [18]. The concept of PRMT5 targeting via competitive inhibition of the substrate has been observed to lead to the clinical development of EPZ015666 and a modified version of EPZ015866 as a treatment for MCL [25,26]. Furthermore, the SAM-competitive PRMT5 inhibitors CMP5 and HLCL61 were developed to suppress PRMT5 enzymatic activity in B-cell lymphoma and AML directly [28,29]. Some PRMT5 inhibitors were verified for anti-tumour effects in HTLV-1-transformed T cells. EPZ015666 resulted in the high toxicity of transformed T cells, including Jurkat, HUT78 (cutaneous T-cell lymphoma cell line), and HTLV-1 infected cell lines compared to resting T cells at a comparable level of IC50 for 12 days. HTLV-1-transformed T cells exhibited a significant increase of apoptosis with the treatment of EPZ015666, but not Jurkat [41]. Treatment of Jurkat and HUT78 cell lines with CMP5 had little effect on cell proliferation and apoptosis; however, the same dose of CMP5 significantly indicated the suppression of cell proliferation in HTLV-1-infected cell lines for 48 h [42], indicating that the inhibition of PRMT5 activity exerted many types of anti-tumour effect at different pathways. We hypothesized that the targeted inhibition of PRMT5 in NDRG2^low^ ATL and solid cancer cells would specifically induce anti-cancer effects as synthetic lethality through the inhibition of cytoplasmic PRMT5/HSP90 activity. Because we examined the anti-tumour functions of PRMT5 inhibitors in primary ATL cells, the detailed mechanism and functions underlying PRMT5 inhibitors need to be investigated using in vivo experiments.

The silencing of PRMT5 or MEP50 expression with each specific shRNA impaired cell proliferation and the stabilization of the HSP90 client proteins AKT and NEMO in NDRG2^low^ ATL and cancer cell lines, but not in NDRG2-expressing cell lines. The PRMT5 inhibitor EPZ015866 was used to examine the effects on ATL cell lines. However, even at a concentration 1000 times higher than the IC50 (nM order) for MCL, highly specific effects such as cell growth inhibition and the degradation of client proteins were not obtained in NDRG2^low^ ATL cell lines. Moreover, CMP5 and HLCL61 were more sensitive to NDRG2^low^ ATL and solid cancer cell lines than NDRG2-expressing cell lines; in addition, they reduced the protein levels of AKT and NEMO through the suppression of HSP90 arginine methylation. Although these PRMT5 inhibitors were identified as first-in-class small molecules of arginine methylation suppression of nuclear histones, they have been shown to exhibit toxicity to normal cells and still require further optimization [28,29]. Because the difference in inhibitory effects of these PRMT5 inhibitors from normal cells is as small as approximately two times compared to NDRG2^low^ tumour cells, we are also encouraged to develop selective and efficient inhibitors targeting phosphorylated PRMT5 or adaptor protein binding sites such as MEP50, rather than protein-substrate and SAM-binding sites. It has been reported that inhibitors of protein–protein interactions (PPIs) between PRMT5 and RioK1, pICIn, or MEP50 can disrupt the complex and selectively reduce substrate methylation, indicating that PPIs support the development and biological characterization of novel PRMT5 inhibitors [43,44,45].

## 4. Materials and Methods

### 4.1. Reagents

The PRMT5 inhibitor EPZ015866 was obtained from Sigma-Aldrich (St. Louis, MO, USA); additionally, CMP5 was obtained from Merck RGaA (Darmstadt, Germany), and HLCL61 was obtained from Cayman Chemical (Ann Arbor, MI, USA). Cell proliferation/cell toxicity kit Cell Counting Kit-8 was purchased from DOJINDO (Kumamoto, Japan). All of the antibodies that were used in this experiment were purchased from the companies listed in Appendix A.

### 4.2. Cell Lines

Jurkat, MOLT4, and MKB1 are HTLV-1-negative human T-ALL cell lines. MT2 and HUT102 are human T-cell lines transformed by HTLV-1 infection. KOB, KK1, and SO4 are IL2-dependent, and SU9T-01 and ED are IL2-independent ATL cell lines. Jurkat, MOLT4 and MKB1 were obtained from the Fujisaki Cell Center, Hayashibara Biochemical Laboratories (Okayama, Japan). MT2 and HUT102 were kind gifts from Dr H. Iha (Oita University, Yufu, Japan) [46,47]. KOB, KK1 and SO4 were kind gifts from Dr Y. Yamada (Nagasaki University, Nagasaki, Japan) [48]. SU9T-01 was a kind gift from Dr N. Arima (Kagoshima University, Kagoshima, Japan) [49]. ED was a kind gift from Dr M. Maeda (Kyoto University, Kyoto, Japan) [50]. Human embryonic kidney cell HEK293GP and oral squamous cell carcinoma cell line SAS were obtained from RIKEN Bioresource Center (Tsukuba, Japan). Osteosarcoma cell line U2OS (HTB-96) was purchased from the American Type Culture Collection (ATCC, Manassas, VA, USA). The procedure for the isolation of mouse embryonic fibroblasts (MEF) from Wild-Type (WT, +/+) and *Ndrg2*-deficient (−/−) mice has been described elsewhere [8]. IL2-dependent ATL cell lines were maintained in RPMI 1640 medium (Nacalai Tesque, Kyoto, Japan) supplemented with 10% fetal bovine serum (FBS) and 10 ng/mL recombinant human IL2 (Peprotech, Rocky Hill, NJ, USA) in a humidified atmosphere of 5% CO_2_ at 37 °C. HTLV-1-negative cell lines, cell lines transformed with HTLV-1 and IL2-independent ATL cell lines were maintained in the same medium without IL2. Other cells were cultured in Dulbecco’s modified Eagle’s medium (DMEM, Nacalai Tesque) supplemented with 10% FBS.

### 4.3. Plasmids

The vector of shRNA against luciferase (shluc-control), human NDRG2 (shNDRG2), human PRMT5 (shPRMT5-3 and 4), PRMT5 constructs, EGFP-tagged PRMT5, the substitution mutant of GR/AA, S335A, S335D, and HA-tagged HSP90A have been previously described [10]. Human MEP50 complementary DNA (cDNA) from MT2 was subcloned into the p3xFLAG-myc-CMV-26 expression vector (Sigma-Aldrich). The shRNA with the different two oligonucleotides DNA sequences against human MEP50 (shMEP50-1 and 2) and mouse PRMT5 (shPRMT5) was cloned into the BamHI–EcoRI site of the RNAi-Ready-pSIREN-RetroQ-ZnGreen vector (Clontech, Mountain View, CA, USA). The sense and antisense shRNA sequences are listed in Appendix A. The transient transfections were performed using polyethylenimine Hydrochloride (PEI-MAX, Polysciences, Warrington, PA, USA) and Amaxa cell line Nucleofector kit V (LONZA, Basel, Switzerland) following the company’s protocol.

### 4.4. Patient Samples

PBMCs obtained from healthy volunteers and patients with ATL were purified by gradient centrifugation. The collection of ATL cells from the patients and CD4+ lymphocytes from volunteers was performed by AutoMACS negative selection using a CD4^+^ T-cell isolation kit (Miltenyi Biotech, Bergisch Gladbach, Germany). The ATL cells were maintained in AIM-V Medium (Thermo Fisher Scientific, Waltham, MA, USA) supplemented with 20% FBS, 10 mM 2-mercaptoethanol (Thermo Fisher Scientific), and 10 ng/mL recombinant human IL2.

### 4.5. Establishment of Stable Knockdown in Cancer Cell Line

The shRNA vectors were co-transfected into 293GP cells along with the envelope plasmid pVSV-G using PEI-MAX reagent according to the manufacturer’s instructions. After six hours of transfection, the medium was changed, and the cells were incubated for 48 h in DMEM supplemented with 10% FBS and 10 μM Forskolin (Sigma-Aldrich). The supernatant containing retrovirus was collected by polyethylene glycol (PEG, Fujifilm Wako Pure Chemical, Osaka, Japan) purification. Two days after retroviral infection in cancer cell lines, ZnGreen-positive cells were sorted with a JSAN cell sorter (Bay Bioscience, Kobe, Japan).

### 4.6. Cell Proliferation Assay and Calculation of IC50

Cells were seeded into 96-well plates at a density of 3 × 10^3^ cells per well and cultured for the indicated time period. Viable cells were counted using Cell Counting Kit-8 according to the manufacturer’s protocol. The values of 50% inhibitory concentration (IC50) were calculated from the following formula: IC50 = 10^[LOG(A/B) × (50 − C)/(D − C) + LOG(B)] − with A, the concentration of the higher side of 50% of absorbance; B, the concentration of the lower side of 50% of absorbance; C, the reduction rate of absorbance at the concentration of B; D, the reduction rate of absorbance at the concentration of A [51].

### 4.7. Western Blot and Immunoprecipitation

Cells were harvested for the extraction of proteins by homogenization in NP-40 lysis buffer (50 mM Tris-HCl, pH 8.0, 150 mM NaCl, 5 mM EDTA, 1% NP-40) supplemented with a proteinase inhibitor cocktail (Sigma-Aldrich) and phosphatase inhibitor tablet (PhosStop, Roche, Basel, Switzerland). The lysates were incubated with 1 μg of the indicated antibodies or normal IgG with constant rotation at 4 °C overnight and were then incubated with Protein G Sepharose 4 Fast Flow (GE Healthcare, Uppsala, Sweden) for 2 h. Equal amounts of protein samples and the immunoprecipitates were boiled at 95 °C for 10 min in 1× SDS sample buffer (62.5 mM Tris-HCl, pH 6.8, 2% SDS, 25% glycerol, 5% 2-mercaptoethanol, 0.01% bromophenol blue), separated by SDS-polyacrylamide gel electrophoresis (SDS-PAGE) and then transferred to a polyvinylidene difluoride membrane (PVDF, Immobilon-P, Millipore, Bedford, MA, USA). The membranes were blocked in TBS (10 mM Tris-HCl, pH 7.4, 100 mM NaCl)–Tween (0.1%) (TBST) with 1% BSA or Blocking One (Nacalai Tesque) and were then probed with primary antibodies diluted in TBST-BSA or Can Get Signal Solution 1 (TOYOBO, Osaka, Japan) overnight in 4 °C. After washing three times with TBST, the membranes were incubated with horseradish peroxidase (HRP)-conjugated secondary antibodies diluted in TBST-BSA or Can Get Signal Solution 2 (TOYOBO) at room temperature for one hour. The bands were detected using a Lumi-light Plus kit (Roche) and LAS-3000 imager (Fujifilm, Tokyo, Japan). The band intensities were quantified with NIH ImageJ software 1.53t. All primary antibodies were used at a dilution of 1:1000.

### 4.8. Statistical Analysis

Data, bars and markers in the figures represent the mean ± s.d. We used the two-tailed Student’s t-test and Mann-Whitney *U*-test for comparisons within each parameter. Statistics were calculated using GraphPad Prism software 9 (GraphPad, San Diego, CA, USA). Differences were considered statistically significant when the *p* value was <0.05.

## 5. Conclusions

Our results may provide a concept of PRMT5-targeting molecules for anti-tumour activity in NDRG2^low^ tumour cells. Furthermore, the decrease in H3R8me2s/H4R3me2s and the increase in HSP90 arginine methylation via the knockdown of NDRG2 expression in NDRG2-expressing T-ALL exhibited sensitivity to CMP5 and HLCL61, along with the degradation of AKT and NEMO. Furthermore, the enhanced expression of NDRG2 suppressed the vulnerability and degradation of client proteins upon treatment with CMP5 and HLCL61 in NDRG2^low^ ATL and solid cancer cell lines, indicating that the targeting of PRMT5/MEP50 enzymatic activity is a feasible and effective strategy for promoting cancer vulnerability in NDRG2^low^ ATL and various cancer cells.

## Figures and Tables

**Figure 1 ijms-25-02842-f001:**
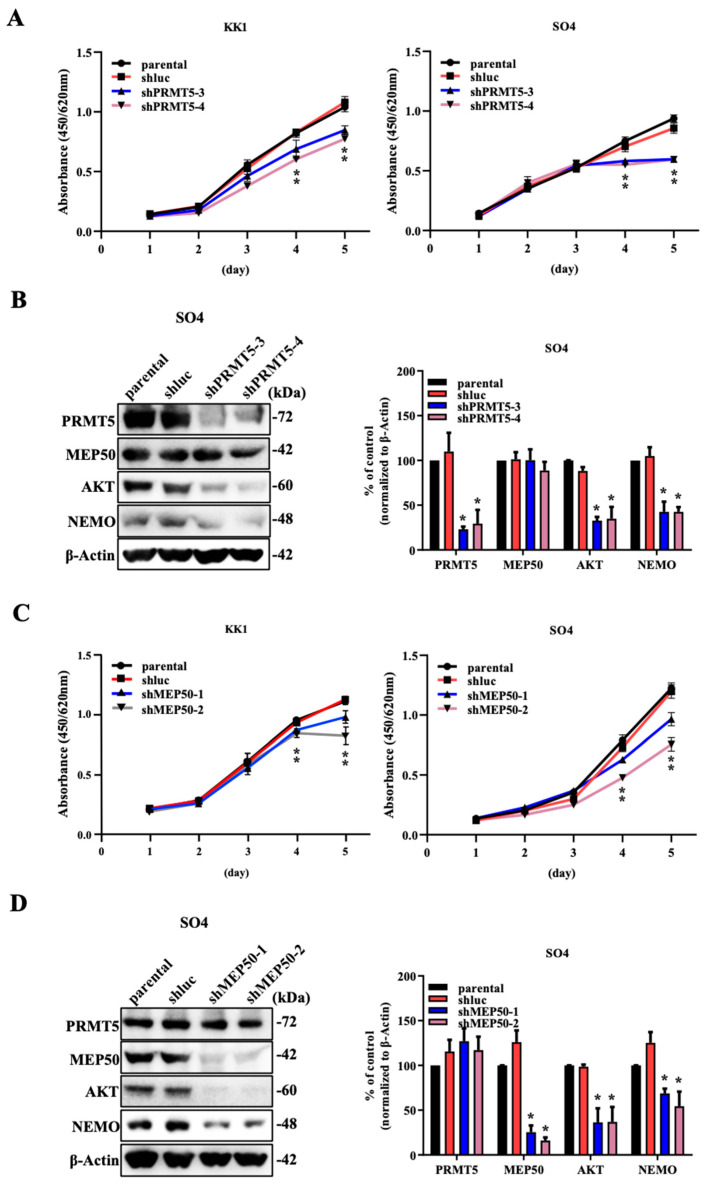
The knockdown of PRMT5/MEP50 expression results in the inhibitory effects on ATL with low NDRG2 expression. (**A**) Cell growth curves of NDRG2^low^ ATL cell lines KK1 and SO4 (parental, shluc, shPRMT5-3, and 4) for five days. The mean and s.d. are shown (*n* = 4), with *: *p* < 0.05 compared with parental. (**B**) Expression of PRMT5, MEP50, AKT, and NEMO in SO4 cells (parental, shluc, shPRMT5-3, and 4) was immunoblotted by each specific antibody. Results are representative of three independent experiments. Bar graphs show the quantification of the relative band intensity normalised to β-actin. The mean and s.d. are shown (*n* = 3), with * *p* < 0.05, compared with parental. (**C**) Cell growth curves of KK1 and SO4 cells (parental, shluc, shMEP50-1, and 2) for five days. The mean and s.d. are shown (*n* = 4), with *: *p* < 0.05 compared with parental. (**D**) Expression of PRMT5, MEP50, AKT, and NEMO in SO4 cells (parental, shluc, shMEP50-1, and 2) was immunoblotted by each specific antibody. Results are representative of three independent experiments. Bar graphs show the quantification of the relative band intensity normalised to β-actin. The mean and s.d. are shown (*n* = 3), with * *p* < 0.05, compared with parental. (**E**) Cell growth curves of NDRG2-expressing T-ALL cell lines Jurkat and MOLT4 (parental, shluc, shPRMT5-3, and 4) for five days. The mean and s.d. are shown (*n* = 4). (**F**) Expression of PRMT5, MEP50, AKT, and NEMO in Jurkat cells (parental, shluc, shPRMT5-3, and 4) was immunoblotted by each specific antibody. Results are representative of three independent experiments. Bar graphs show the quantification of the relative band intensity normalised to β-actin. The mean and s.d. are shown (*n* = 3), with * *p* < 0.05, compared with parental. (**G**) Cell growth curves of Jurkat and MOLT4 cells (parental, shluc, shMEP50-1, and 2) for five days. The mean and s.d. are shown (*n* = 4), with *: *p* < 0.05 compared with parental. (**H**) Expression of PRMT5, MEP50, AKT, and NEMO in Jurkat cells (parental, shluc, shMEP50-1, and 2) were immunoblotted by each specific antibody. Results are representative of three independent experiments. Bar graphs show the quantification of the relative band intensity normalised to β-actin. The mean and s.d. are shown (*n* = 3), with * *p* < 0.05, compared with parental.

**Figure 2 ijms-25-02842-f002:**
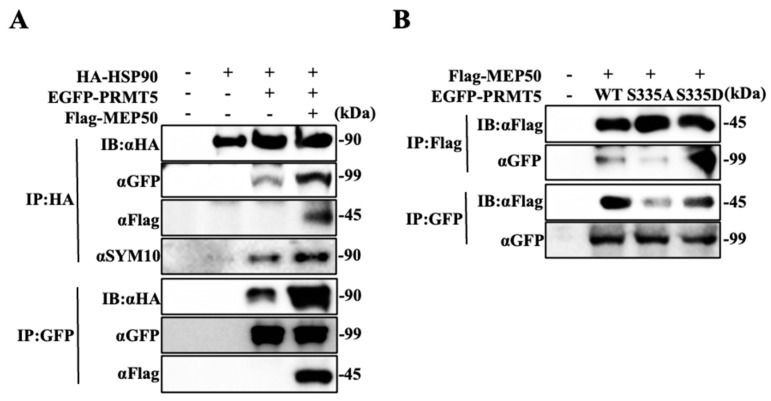
Hyperphosphorylated PRMT5 binds to MEP50 and promotes HSP90 arginine methylation. (**A**) The lysates of 293T cells transfected with HA-tagged HSP90, EGFP-tagged PRMT5, and/or Flag-tagged MEP50 were immunoprecipitated with anti-HA or anti-GFP antibody, and immunoprecipitates were assayed by western blotting with the indicated antibodies. (**B**) EGFP-tagged wild-type or mutated PRMT5 (S335A and S335D) was co-transfected with Flag-tagged MEP50. The cell lysates were immunoprecipitated with anti-GFP or anti-Flag antibodies and subsequently immunoblotted with each indicated antibody.

**Figure 3 ijms-25-02842-f003:**
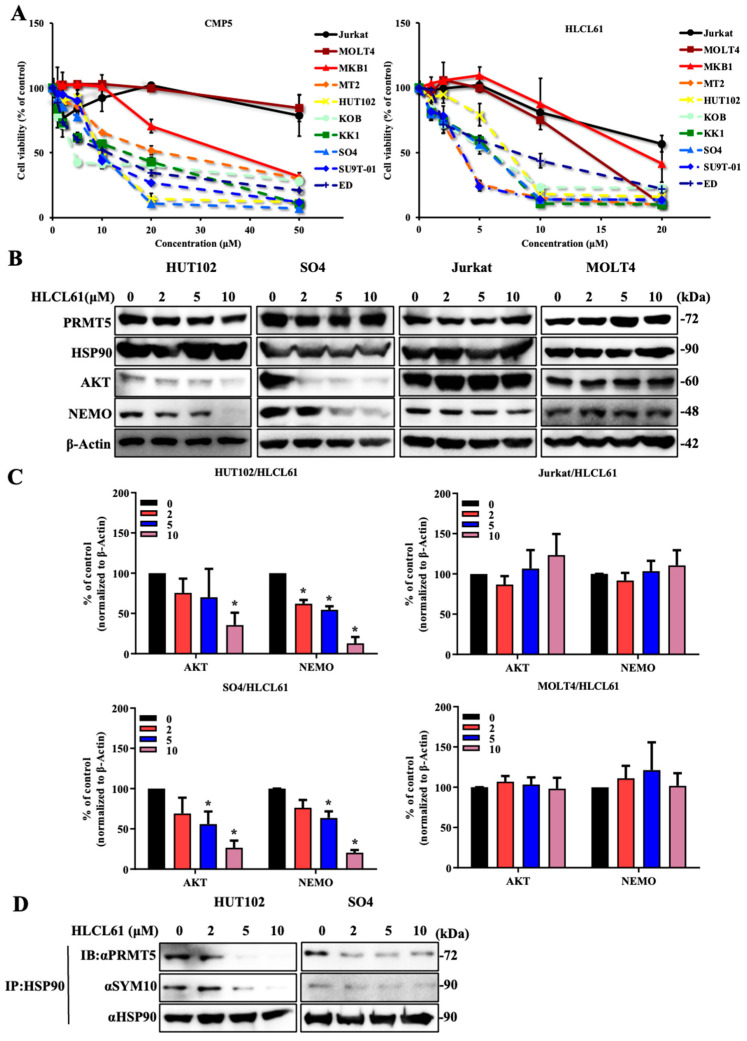
NDRG2^low^ ATL cells are sensitive to PRMT5 inhibitors. (**A**) Cell viability and IC50 were determined after treatment with 0–50 μM CMP5 and 0–20 μM HLCL61 at 120 h in T-ALL, HTLV-1-infected, and ATL cell lines. (**B**) Expression of PRMT5, HSP90, AKT, and NEMO in HUT102, SO4, Jurkat, and MOLT4 cells after treatment with the indicated doses of HLCL61 for 24 h was determined via immunoblot analysis using each specific antibody. Results are representative of three independent experiments. (**C**) Bar graphs show the quantification of the relative band intensity normalised to β-actin. The mean and s.d. are shown (*n* = 3), with * *p* < 0.05, compared with 0. (**D**) HSP90 immunoprecipitated from HUT102 and SO4 cells treated with the indicated doses of HLCL61 for 24 h was analysed via immunoblotting using the indicated antibodies.

**Figure 4 ijms-25-02842-f004:**
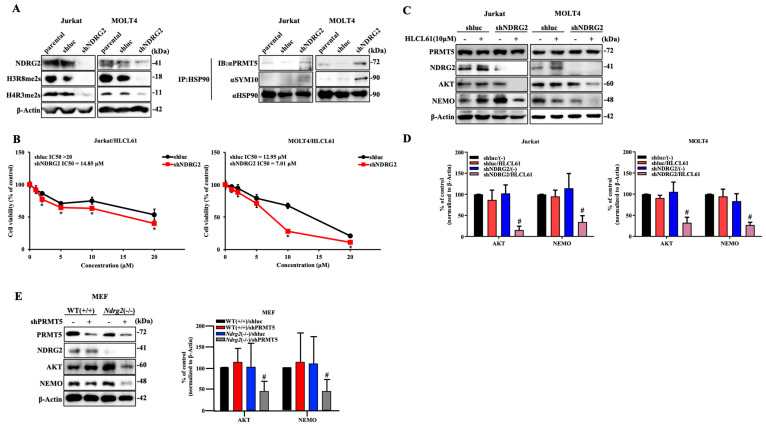
Knockdown of NDRG2 expression in T-ALL cells enhances sensitivity to PRMT5 inhibitors. (**A**) Expression of NDRG2, H3R8me2s, and H4R3me2s in Jurkat and MOLT4 cells (parental, shluc, and shNDRG2) was immunoblotted using each specific antibody. HSP90 immunoprecipitated from Jurkat and MOLT4 cells was analysed via immunoblotting using the indicated antibodies. (**B**) Cell viability and IC50 were determined after treatment with 0–20 μM HLCL61 at 120 h in Jurkat and MOLT4 cells (shluc and shNDRG2). The mean and s.d. are shown (*n* = 4), with *: *p* < 0.05 compared with shluc. (**C**) Expression of PRMT5, NDRG2, AKT, and NEMO in Jurkat and MOLT4 cells (shluc and shNDRG2) after treatment with 10 μM HLCL61 for 24 h was determined via immunoblot analysis using each specific antibody. Results are representative of three independent experiments. (**D**) Bar graphs show the quantification of the relative band intensity normalised to β-actin. The mean and s.d. are shown (*n* = 3), with # *p* < 0.05, compared with shluc/HLCL61. (**E**) Whole-cell lysates from MEFs of WT(+/+) and *Ndrg2*(−/−) mice after PRMT5 knockdown were subjected to western blotting using the indicated antibodies. Results are representative of three independent experiments. Bar graphs show the quantification of the relative band intensity normalised to β-actin. The mean and s.d. are shown (*n* = 3), with # *p* < 0.05, compared with WT(+/+)/shPRMT5.

**Figure 5 ijms-25-02842-f005:**
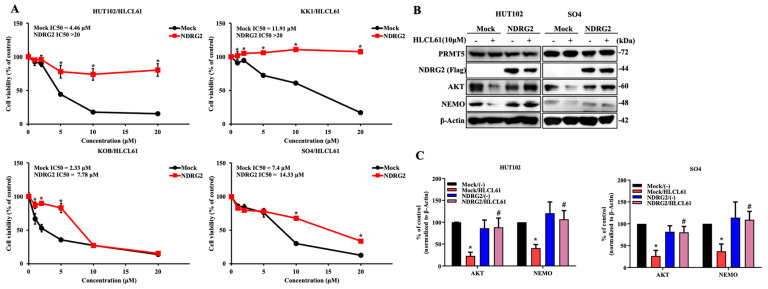
The enhanced expression of NDRG2 attenuates the anti-tumour effects of PRMT5 inhibitors in ATL and solid cancer cells. (**A**) Cell viability and IC50 were determined after treatment with 0–20 μM HLCL61 at 120 h in HUT102 and SO4 cells (Mock and NDRG2). The mean and s.d. are shown (*n* = 4), with *: *p* < 0.05 compared with Mock. (**B**) Expression of PRMT5, NDRG2 (Flag), AKT, and NEMO in HUT102, KOB, KK1, and SO4 cells (Mock and NDRG2) after treatment with 10 μM HLCL61 for 24 h was determined via immunoblot analysis. Results are representative of three independent experiments. (**C**) Bar graphs show the quantification of the relative band intensity normalised to β-actin. The mean and s.d. are shown (*n* = 3), with * *p* < 0.05, compared with Mock/(-) and # *p* < 0.05, compared with Mock/HLCL61.

**Figure 6 ijms-25-02842-f006:**
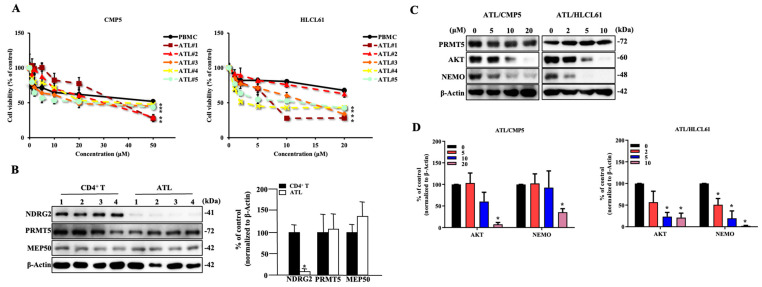
PRMT5 inhibitors are effective against NDRG2^low^ ATL patient cells. (**A**) Cell viability and IC50 were determined after treatment with 0–50 μM CMP5 and 0–20 μM HLCL61 at 120 h in PBMCs and patient-derived ATL cells. The mean and s.d. are shown (*n* = 4), with *: *p* < 0.05 compared to PBMCs. (**B**) Western blot analysis of NDRG2, PRMT5, and MEP50 was performed in the CD4^+^ T-cells from healthy volunteers (CD4^+^ T) served as the controls and patient-derived ATL cells. Bar graphs show the quantification of the relative band intensity normalised by β-actin in CD4^+^ T and ATL. The mean and s.d. are shown (*n* = 4), with * *p* < 0.05, compared to CD4^+^ T. (**C**) Expression of PRMT5, AKT, and NEMO in ATL cells after treatment with the indicated doses of CMP5 and HLCL61 for 24 h was determined via immunoblot analysis. Results are representative of three independent experiments. (**D**) Bar graphs show the quantification of the relative band intensity normalised to β-actin. The mean and s.d. are shown (*n* = 3), with * *p* < 0.05, compared with 0.

**Figure 7 ijms-25-02842-f007:**
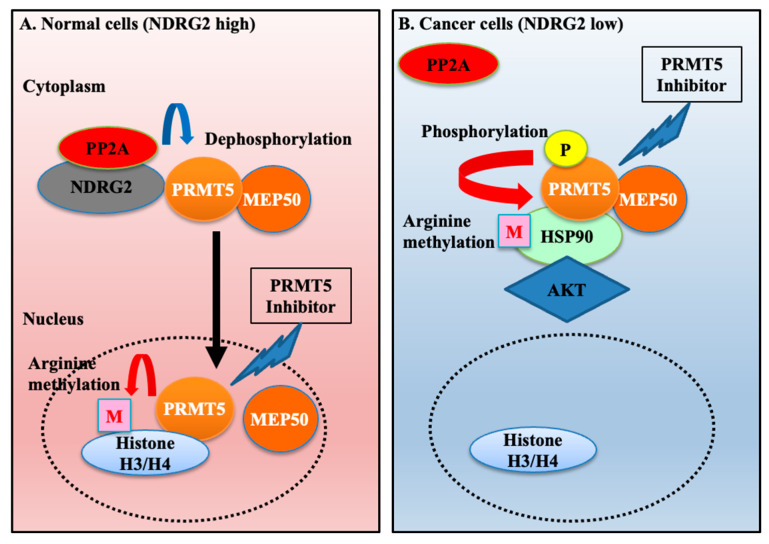
Scheme of cancer vulnerability between the downregulation of NDRG2 and cytoplasmic PRMT5/MEP50 activity in NDRG2^low^ cancer cells. (**A**) Hypophosphorylated PRMT5 via the recruitment of NDRG2/PP2A is translocated to the nucleus, leading to the induction of histone arginine methylation in NDRG2-expressing cells to participate in gene expression and several physiological processes. Therefore, treatment of PRMT5-specific inhibitors does not induce anti-cancer effects. (**B**) Highly phosphorylated PRMT5 firmly associates with MEP50 and HSP90 in NDRG2^low^ cancer cells, leading to the enhancement of HSP90 arginine methylation and chaperone activity. Thus, interference with cytoplasmic PRMT5/MEP50 activity causes cancer vulnerability, followed by the inhibition of tumorigenesis.

**Table 1 ijms-25-02842-t001:** Inhibitory effect of various PRMT5 inhibitors on cell proliferation of non-ATL T-ALL and ATL-related cells at 120 h. The numbers represent IC50 (μM).

Cell Lines	EPZ015866	CMP5	HLCL61	Tax	NDRG2
Jurkat	>100	71.7	22.72	-	+
MOLT4	>100	92.97	13.06	-	+
MKB1	>100	32.5	17.6	-	+
MT2	>100	21.65	3.09	+	-
HUT102	>100	9.49	6.95	+	-
KOB	>100	3.98	5.52	+	-
SU9T-01	>100	9.1	3.23	+	-
KK1	>100	14.07	5.7	-	-
SO4	>100	9.45	5.57	-	-
ED	>100	10.81	7.58	-	-

**Table 2 ijms-25-02842-t002:** Inhibitory effect of PRMT5 inhibitors on cell proliferation of leukemia cells from ATL patients at 120 h. The numbers represent IC50 (μM). PBMC indicates mononuclear cells in the peripheral blood.

Cell Origin	CMP5	HLCL61
PBMC	58.08	43.37
ATL#1	33.12	6.97
ATL#2	27.13	42.71
ATL#3	18.81	28.46
ATL#4	35	2.33
ATL#5	23.94	12.06

## Data Availability

Data are contained within the article and Appendix A.

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
