# Peer review of "Inhibition of PRMT5/MEP50 Arginine Methyltransferase Activity Causes Cancer Vulnerability in NDRG2low Adult T-Cell Leukemia/Lymphoma"

_ijms, 2024, doi:10.3390/ijms25052842_

Round 1

Reviewer 1 Report

Comments and Suggestions for Authors

Major comments:

1.  The abstract of the manuscript does not reflect the results presented. A rewrite is recommended.

2. The Introduction section does not present any background knowledge necessary for interpretation. It lacks some basic references on this topic, which describe the phosphorylation of PRMT5 regulating the same processes and is relevant to the results presented in this article, but its comparison with them is also important. (10.3390/ijms21103684; 10.1038/srep40590 ).

3. The authors of the manuscript claim that the effect of PRMT in the system they use is not the result of gene expression, but of post-translational modification. Based on the results presented (many supporting experiments are lacking), it is rather gene expression that indirectly affects AKT/Hsp90 proteins. I recommend a review of the existing PRMT5 KO database to verify whether silencing has affected elements of the putative signaling pathway. Based on only one blot shown (the series shown in the Supplementary, is the same as in the figures), the assumption is doubtable. The conclusions can be drawn only by the application of PP2A or hsp90  inhibitors.

4.. Its purification will involve MEP50 both endogenously and exogenously.

5. Again, only one series of blots is presented (even in Supplementary). At least 3 are necessary.  The Flag IP experiment is required for drawing any conclusions.

6. The manuscript provides no information about the potential effect of PRMT5 on the expression of NDRG2. That should be verified and presented in Fig. 1.

7. There is not a single sign of any localization or fractionation experiments to prove the PRMT5 translocation that the authors suggest in Fig. 7.

8. Any proof that the PRMT5 inhibitors have no effect on MEP50 binding and only the methyl donor site is affected not the substrate site?

Minor comments:

1.     The quality of the figures are rather poor in many cases. Technically, it is not acceptable to calculate with any density of the blots beyond the linear phase of the detection. Recalculate the semi-quantitative WBs with a lower exposure time for AKT, beta-actin, and eHSP90. The description of the calculation of these data is missing.

2.     For Western blot analysis, the molecular weight markers must be presented on the figure everywhere (f.i. Fig. 2).

3.     Figure 1A is not visible. I recommend another visualization of the results. The labels are not visible at all.

4.     The dataset is questionable since only one version of the original blots is presented for Fig 1. How is it possible to calculate any statistic with 1 data?

5.     It is not clear in the Mat and meth that the data presented are biological or technical parallel experiments.

6.     It is rather unusual that the entire Material and Methods section is presented as Supplementary. I highly advise to move it to the main text or use citations and make it shorter but it should certainly go to the main text in some form.

Comments on the Quality of English Language

Minor revision of the Abstract and Introduction is required.

Author Response

We are most grateful to the reviewers for their helpful comments on the original version of our manuscript. We have taken all these comments into account and submitted a revised version of our paper.

We have addressed the comments by reviewers and we hope that our explanations and revisions are satisfactory.

We hope that the revised version of our paper is now suitable for publication in International Journal of Molecular Sciences,and we look forward to hearing from you at your earliest convenience.

Reviewer: 1

Comments and Suggestions for Authors

Major comments:

  1. The abstract of the manuscript does not reflect the results presented. A rewrite is recommended.

As recommended by the reviewer, we have changed the abstract (lines 29-39, Red highlight).

  1. The Introduction section does not present any background knowledge necessary for interpretation. It lacks some basic references on this topic, which describe the phosphorylation of PRMT5 regulating the same processes and is relevant to the results presented in this article, but its comparison with them is also important. (10.3390/ijms21103684; 10.1038/srep40590).

In response to the reviewer's comments, we have added some sentences in the Introduction section (lines 69-76, Red highlight) and in the Reference section (11, 12, Red highlight).

  1. The authors of the manuscript claim that the effect of PRMT in the system they use is not the result of gene expression, but of post-translational modification. Based on the results presented (many supporting experiments are lacking), it is rather gene expression that indirectly affects AKT/Hsp90 proteins. I recommend a review of the existing PRMT5 KO database to verify whether silencing has affected elements of the putative signaling pathway. Based on only one blot shown (the series shown in the Supplementary, is the same as in the figures), the assumption is doubtable. The conclusions can be drawn only by the application of PP2A or hsp90 inhibitors.

As recommended by the reviewer, we have checked the review paper as follows (1). Gene regulation by PRMT5 is thought to lead to physiological functions through the arginine methylation of histones, and transcription factors, such as p53, E2F1, and p65 in the nucleus (1). However, as shown in our previous paper (2), when NDRG2 expression is suppressed in ATL cells, PRMT5 localizes from the nucleus to the cytoplasm, resulting in the reduction of arginine methylated nuclear proteins and gene regulation functions. Thus, as we have reported, the paper is primarily concerned with the enhancement of HSP90 by arginine methylation as the primary function of PRMT5 in the cytoplasm. Moreover, other paper identified PRMT5 substrates and performed methylome profiling after PRMT5 knockdown (KO) in acute myeloid leukemia (AML). 2962 proteins were differentially expressed in PRMT5 KO and control, and enriched in DNA replication/repair, RNA processing, and cytosol proteins. In addition, potential PRMT5 substrates were enriched in proteins involved in RNA end processing, splicing, and protein binding, including HSP90 (3). We suggest that PRMT5 may be involved in cytosolic proteins through the regulation of HSP90 functions. Furthermore, it has been reported that HSP90 exhibited the direct interaction with PRMT5 and the anti-tumor function of HSP90 inhibitor was regulated by PRMT5 expression in cancer cells (4, 5). We confirmed that the treatment with HSP90 inhibitor 17-AAG was inhibited cell proliferation of ATL cell lines with the degradation of PRMT5 and signaling molecules (AKT and NEMO), suggesting that PRMT5 maintains the AKT and NEMO proteins via the post-translational modification (arginine methylation) of HSP90 in ATL (2).

  1. Zhu F, Rui L. PRMT5 in gene regulation and hematologic malignancies. Genes Dis. 2019 Jun 19;6(3):247-257. doi: 10.1016/j.gendis.2019.06.002. PMID: 32042864; PMCID: PMC6997592.
  2. Ichikawa T, Shanab O, Nakahata S, Shimosaki S, Manachai N, Ono M, Iha H, Shimoda K, Morishita K. Novel PRMT5-mediated arginine methylations of HSP90A are essential for maintenance of HSP90A function in NDRG2lowATL and various cancer cells. Biochim Biophys Acta Mol Cell Res. 2020 Feb;1867(2):118615. doi: 10.1016/j.bbamcr.2019.118615. Epub 2019 Nov 22. PMID: 31765670.
  3. Radzisheuskaya A, Shliaha PV, Grinev V, Lorenzini E, Kovalchuk S, Shlyueva D, Gorshkov V, Hendrickson RC, Jensen ON, Helin K. PRMT5 methylome profiling uncovers a direct link to splicing regulation in acute myeloid leukemia. Nat Struct Mol Biol. 2019 Nov;26(11):999-1012. doi: 10.1038/s41594-019-0313-z. Epub 2019 Oct 14. PMID: 31611688; PMCID: PMC6858565.
  4. Maloney A, Clarke PA, Naaby-Hansen S, Stein R, Koopman JO, Akpan A, Yang A, Zvelebil M, Cramer R, Stimson L, Aherne W, Banerji U, Judson I, Sharp S, Powers M, deBilly E, Salmons J, Walton M, Burlingame A, Waterfield M, Workman P. Gene and protein expression profiling of human ovarian cancer cells treated with the heat shock protein 90 inhibitor 17-allylamino-17-demethoxygeldanamycin. Cancer Res. 2007 Apr 1;67(7):3239-53. doi: 10.1158/0008-5472.CAN-06-2968. PMID: 17409432.
  5. Sharp SY, Prodromou C, Boxall K, Powers MV, Holmes JL, Box G, Matthews TP, Cheung KM, Kalusa A, James K, Hayes A, Hardcastle A, Dymock B, Brough PA, Barril X, Cansfield JE, Wright L, Surgenor A, Foloppe N, Hubbard RE, Aherne W, Pearl L, Jones K, McDonald E, Raynaud F, Eccles S, Drysdale M, Workman P. Inhibition of the heat shock protein 90 molecular chaperone in vitro and in vivo by novel, synthetic, potent resorcinylic pyrazole/isoxazole amide analogues. Mol Cancer Ther. 2007 Apr;6(4):1198-211. doi: 10.1158/1535-7163.MCT-07-0149. PMID: 17431102.

  1. Its purification will involve MEP50 both endogenously and exogenously.

As expected by the reviewer, MEP50 binds and functions with PRMT5, whether endogenously or exogenously introduced PRMT5. We agree with the reviewer’s comment.

  1. Again, only one series of blots is presented (even in Supplementary). At least 3 are necessary.  The Flag IP experiment is required for drawing any conclusions.

We thank the reviewer for pointing this out. The results are representative of three independent experiments. Bar graphs show the quantification of the relative signal intensity normalized to β-actin. Data are expressed as the mean and s.d.(n=3).

We have added some sentences to the corresponding figure legends (Red highlight).

  1. The manuscript provides no information about the potential effect of PRMT5 on the expression of NDRG2. That should be verified and presented in Fig. 1.

Previous results (Biochim Biophys Acta Mol Cell Res. 2020) indicated that the expression of PRMT5 and MEP50 was unchanged in NDRG2-enhanced ATL cell lines compared with control ATL cell lines. Furthermore, the protein expression level of endogenous and exogenous NDRG2 was unchanged in T-ALL and ATL cell lines with PRMT5 inhibitor treatment in Figures 4C and 5B. We hope that the reviewer is satisfied with these explanations and responses.

  1. There is not a single sign of any localization or fractionation experiments to prove the PRMT5 translocation that the authors suggest in Fig. 7.

Previous results (Biochim Biophys Acta Mol Cell Res. 2020) indicated that PRMT5 was detected in the nucleus with histone arginine methylation (H3R8me2s) in NDRG2-expressing T-ALL cells (MOLT4). On the other hands, PRMT5 in NDRG2lowATL cells (HUT102), was mainly resided in the cytoplasm with extremely low histone arginine methylation and formed a complex with HSP90 with high methylation. Furthermore, when we investigated the phosphorylation level of PRMT5 in ATL cells by nano-LC/MS analysis, PRMT5 was highly phosphorylated at Serine 335 (S335). The S335 phosphorylation of PRMT5 in ATL cells was disappeared by enforced NDRG2 expression, suggesting that PRMT5 is mainly localized in the cytoplasm with high S355 phosphorylation in NDRG2low ATL cells.

  1. Any proof that the PRMT5 inhibitors have no effect on MEP50 binding and only the methyl donor site is affected not the substrate site?

As the SAM-competitive PRMT5 inhibitors CMP5 and HLCL61 were developed to directly inhibit PRMT5 enzymatic activity in tumours, we have not had information on whether PRMT5 disassociates with MEP50 after the treatment with PRMT5 inhibitors.  We are focusing on the analysis and development of selective and efficient inhibitors targeting the PRMT5/MEP50 interaction.

Minor comments:

  1. The quality of the figures are rather poor in many cases. Technically, it is not acceptable to calculate with any density of the blots beyond the linear phase of the detection. Recalculate the semi-quantitative WBs with a lower exposure time for AKT, beta-actin, and eHSP90. The description of the calculation of these data is missing.

As responded above, we performed three independent experiments. The data are representative of three independent experiments. Bar graphs show quantification of the relative band intensity normalized to β-actin. Data are expressed as the mean and s.d. (n=3).

  1. For Western blot analysis, the molecular weight markers must be presented on the figure everywhere (f.i. Fig. 2).

In response to the reviewer's comments, we added molecular weight markers to Figure 2 (Red highlight).

  1. Figure 1A is not visible. I recommend another visualization of the results. The labels are not visible at all.

To address reviewer’s comment, we made the figure size bigger for the reviewer to visualize.

  1. The dataset is questionable since only one version of the original blots is presented for Fig 1. How is it possible to calculate any statistic with 1 data?

The data are representative of three independent experiments. Bar graphs show the quantification of the relative band intensity normalized to β-actin. Data are expressed as the mean and s.d. (n=3).

  1. It is not clear in the Mat and meth that the data presented are biological or technical parallel experiments.

Data, bars, and markers in the figures represent the mean ± s.d. We used the two-tailed Student’s t-test and Mann-Whitney U-test for comparisons within each parameter. Statistics were calculated using GraphPad Prism software (GraphPad, San Diego, CA, USA). Differences were considered statistically significant when the P value was <0.05. This passage was added to the Materials and Methods section (lines 566-569, Red highlight).

  1. It is rather unusual that the entire Material and Methods section is presented as Supplementary. I highly advise to move it to the main text or use citations and make it shorter but it should certainly go to the main text in some form.

To address the reviewer’s comment, we have moved “Supplementary Materials and methods” to “main text”. We have modified some sentences in the Materials and Methods (lines 479-569, Red highlight) and the References sections (46-51, Red highlight).

Reviewer 2 Report

Comments and Suggestions for Authors

Adult T-cell leukemia/lymphoma (ATL) is a highly aggressive malignancy caused by infection with the oncogenic retrovirus human T-cell leukemia virus type 1 (HTLV-1). The viral protein Tax is a powerful oncoprotein that plays a crucial role in the pathogenesis of HTLV-I-associated diseases including ATL. N-myc downstream-regulated gene 2 (NDRG2), which is a tumor suppressor, is frequently lost in many types of tumors, including ATL. They used several cancer cell lines, including ATL, and primary ATL cells to test whether the inhibition of PRMT5 activity is a drug target in NDRG2low tumors. In addition, they employed knockdown experiments of PRMT5 and binding partner methylsome protein 50 (MEP50) as well as the use of the SAM-competitive PRMT5-specific inhibitors CMP5 and HLCL61.

The authors showed interesting results regarding the knockdown of PRMT5/MEP50 expression results in the inhibition of cell proliferation through the degradation of specific proteins in ATL and various cancer cells with low NDRG2; NDRG2low ATL cells are sensitive to PRMT5 inhibitors; knockdown of NDRG2 expression in T-ALL cells enhances sensitivity to PRMT5 inhibitors; The enhanced expression of NDRG2 attenuates the antitumor effects of PRMT5 inhibitors in ATL and solid cancer cells; and finally PRMT5 inhibitors are effective against NDRG2low ATL patient cells.

These studies are interesting, well-written, and referenced. I would rather that the authors focus in this manuscript on ATL and HTLV-I negative T lymphoma cells. The authors have previously published in BBA-2019 about the regulation of NDRG2 expression during ATL development after HTLV-1 infection. What is this study lacking is the effect of different treatments on tax levels in ATL cells and where tax plays a role in their scheme in Figure 7.

The authors should present the different levels of tax and NDRG2 in the ATL cell lines presented in table 2 and primary ATL cells.

In vivo studies would strengthen their manuscript and should be performed or at least discussed in this manuscript.

Table 1 and 2- The authors should mention at what time point were the IC50 calculated.

Author Response

We are most grateful to reviewers for their helpful comments on the original version of our manuscript. We have taken all these comments into account and submitted a revised version of our paper.

We have addressed the comments by reviewers and we hope that our explanations and revisions are satisfactory.

We hope that the revised version of our paper is now suitable for publication in International Journal of Molecular Sciences,and we look forward to hearing from you at your earliest convenience.

Reviewer: 2

Comments and Suggestions for Authors

Adult T-cell leukemia/lymphoma (ATL) is a highly aggressive malignancy caused by infection with the oncogenic retrovirus human T-cell leukemia virus type 1 (HTLV-1). The viral protein Tax is a powerful oncoprotein that plays a crucial role in the pathogenesis of HTLV-I-associated diseases including ATL. N-myc downstream-regulated gene 2 (NDRG2), which is a tumor suppressor, is frequently lost in many types of tumors, including ATL. They used several cancer cell lines, including ATL, and primary ATL cells to test whether the inhibition of PRMT5 activity is a drug target in NDRG2low tumors. In addition, they employed knockdown experiments of PRMT5 and binding partner methylsome protein 50 (MEP50) as well as the use of the SAM-competitive PRMT5-specific inhibitors CMP5 and HLCL61. 

The authors showed interesting results regarding the knockdown of PRMT5/MEP50 expression results in the inhibition of cell proliferation through the degradation of specific proteins in ATL and various cancer cells with low NDRG2; NDRG2low ATL cells are sensitive to PRMT5 inhibitors; knockdown of NDRG2 expression in T-ALL cells enhances sensitivity to PRMT5 inhibitors; The enhanced expression of NDRG2 attenuates the antitumor effects of PRMT5 inhibitors in ATL and solid cancer cells; and finally PRMT5 inhibitors are effective against NDRG2low ATL patient cells.

These studies are interesting, well-written, and referenced. I would rather that the authors focus in this manuscript on ATL and HTLV-I negative T lymphoma cells. The authors have previously published in BBA-2019 about the regulation of NDRG2 expression during ATL development after HTLV-1 infection. What is this study lacking is the effect of different treatments on tax levels in ATL cells and where tax plays a role in their scheme in Figure 7.

We examined the anti-tumour effects of PRMT5 inhibitors in Tax-positive HTLV-1-infected (HUT102), Tax-positive ATL (KOB), and Tax-negative ATL (SO4 and KK1) cells. Tax expression may not be involved in PRMT5 functions in HTLV-1-infected cells.

The authors should present the different levels of tax and NDRG2 in the ATL cell lines presented in table 2 and primary ATL cells.

The PRMT5 inhibitors were administered to Tax-negative T-ALL (Jurkat, MOLT4, and MKB1), Tax-positive HTLV-1 infected (MT2 and HUT102), Tax-positive ATL (KOB and SU9T-01), Tax-negative ATL (KK1, SO4, and ED) cells, and Tax-negative primary ATL cells. Furthermore, NDRG2 expression was remarkably suppressed in HTLV-1 infected, ATL, and primary ATL cells.

We added expression levels of Tax and NDRG2 to Table 1 as indicated +/-, and some sentences in the Results section (lines 238-239, Blue highlight).

In vivo studies would strengthen their manuscript and should be performed or at least discussed in this manuscript.

In response to the reviewer's comments, we have added some sentences in the Discussion section (lines 446-449, Blue highlight).

Table 1 and 2- The authors should mention at what time point were the IC50 calculated.

In response to the reviewer's comments, we have added the time point (120 h) to the text of the Results section and to Tables 1 and 2 (Blue highlight).

Round 2

Reviewer 2 Report

Comments and Suggestions for Authors

The authors have adequately answered my concerns and comments.